# Synthesis of Titanium-Based Powders from Titanium Oxy-Sulfate Using Ultrasonic Spray Pyrolysis Method

**DOI:** 10.3390/ma17194779

**Published:** 2024-09-28

**Authors:** Duško Kostić, Srecko Stopic, Monika Keutmann, Elif Emil-Kaya, Tatjana Volkov Husovic, Mitar Perušić, Bernd Friedrich

**Affiliations:** 1IME Process Metallurgy and Metal Recycling, RWTH Aachen University, 52056 Aachen, Germany; dkostic@metallurgie.rwth-aachen.de (D.K.); mkeutmann@metallurgie.rwth-aachen.de (M.K.); bfriedrich@metallurgie.rwth-aachen.de (B.F.); 2Faculty of Technology Zvornik, University of East Sarajevo, Karakaj 34A, 75400 Zvornik, Republic of Srpska, Bosnia and Herzegovina; mitar.perusic@tfzv.ues.rs.ba; 3Department of Materials Science and Engineering, Norwegian University of Science and Technology, Høgskoleringen 1, 7034 Trondheim, Norway; elif.e.kaya@ntnu.no; 4Faculty of Technology and Metallurgy, University of Belgrade, Karnegijeva 4, 11000 Belgrade, Serbia; tatjana@tmf.bg.ac.rs

**Keywords:** ultrasonic spray pyrolysis, titanium-based powders, nanoparticles

## Abstract

Submicron and nanosized powders have gained significant attention in recent decades due to their broad applicability in various fields. This work focuses on ultrasonic spray pyrolysis, an efficient and flexible method that employs an aerosol process to synthesize titanium-based nanoparticles by transforming titanium oxy-sulfate. Various parameters are monitored to better optimize the process and obtain better results. Taking that into account, the influence of temperature on the transformation of titanium oxy-sulfate was monitored between 700 and 1000 °C. In addition to the temperature, the concentration of the starting solution was also changed, and the flow of hydrogen and argon was studied. The obtained titanium-based powders had spherical morphology with different particle sizes, from nanometer to submicron, depending on the influence of reaction parameters. The control of the oxygen content during synthesis is significant in determining the structure of the final powder.

## 1. Introduction

Spray pyrolysis is a flexible technique used for producing single- and multi-layered films, dense or porous ceramic coatings, and various material powders. Spray processing methods can be categorized by the energy source that initiates the reaction in the starting solution, including tubular reactors, emulsion combustion, vapor flame reactors, and flame spray pyrolysis. Additionally, these methods can be classified by the atomization technique used for the precursors, such as electrostatic, air-pressurized, and ultrasonic spray pyrolysis [1,2,3,4].

Spray pyrolysis techniques can also be categorized based on the type of atomizer used. The size of the aerosol droplets, which directly impacts the quality of the resulting film, is largely determined by the atomization method. The three primary atomization methods are electrostatic, air blast, and ultrasonic [5,6,7,8].

Ultrasonic atomization functions through an electromechanical device vibrating at high frequencies. This technique is most effective for Newtonian fluids with low viscosity. As the fluid moves across the vibrating surface, the high-frequency vibrations cause it to break up into fine droplets [9,10,11,12,13].

The ultrasonic spray pyrolysis (USP) technique generates droplets using ultrasonic waves, offering several advantages such as simplicity, cost-effectiveness, continuous operation, high deposition rates, and the ability to cover large surface areas. The average size of the droplets produced is typically less than 20 μm at low in-flight speeds [14]. This technique enables the synthesis of metal, oxide, and composite nanomaterials, providing precise control over their shapes and chemical compositions by utilizing metal salts in aqueous solutions [15,16].

USP can be used not only to produce nanoparticles from their precursors but also for the subsequent reduction of precursors to metals in a hydrogen atmosphere. This method is very efficient in the synthesis of fine spherical metal nanoparticles [7,17,18,19,20,21,22].

Titanium dioxide (TiO_2_) nanopowder has become an important material in various fields due to its unique properties, including high refractive index, strong UV absorption, chemical stability, and photocatalytic activity. Its applications span across multiple industries, including environmental remediation, energy, health, and consumer products.

One of the most significant uses of TiO_2_ nanopowder is in photocatalysis. In the energy sector, TiO_2_ nanopowder is utilized in the development of dye-sensitized solar cells (DSSCs). TiO_2_ nanopowder is also extensively used in the production of sunscreens due to its effective UV-blocking capabilities. In the medical field, TiO_2_ nanopowder is explored for its potential in drug delivery systems, antimicrobial coatings, and cancer treatment. Furthermore, TiO_2_ nanopowder is used as a pigment in paints, coatings, and plastics due to its high opacity and brightness. In the field of electronics, TiO_2_ nanoparticles are being studied for their potential in resistive switching devices, which are the basis of next-generation memory technologies [23,24,25,26,27].

Titanium dioxide (TiO_2_) nanopowder production involves several methods, each tailored to achieve specific particle sizes, shapes, and purity levels. These methods include sol–gel processes, hydrothermal synthesis, electrochemical deposition, and physical methods like mechanical milling. The selection of the production method is determined by the intended application, as different processes can result in TiO_2_ nanoparticles with distinct properties [28,29,30].

The main aim of this work is the synthesis of titanium-based powders from titanium oxy-sulfate using the USP method, which is missing in the literature. The preparation of titanium and titanium oxide with the formation of sulfur trioxide and decreasing oxygen content was tested across a temperature range (700–1000 °C).

Ultrasonic spray pyrolysis (USP) is a highly effective and versatile technique for synthesizing titanium dioxide (TiO_2_) nanoparticles. One of its key advantages is the precise control over morphology. By generating a fine aerosol mist of the precursor solution through ultrasonic waves, the droplet size can be adjusted by altering the ultrasonic frequency, resulting in uniform, well-defined nanoparticles after thermal decomposition. Additionally, USP is easily scalable, enabling large-scale production without compromising nanoparticle quality, making it ideal for industrial applications. Furthermore, the method allows for the synthesis of high-purity TiO_2_ nanoparticles.

## 2. Materials and Methods

Ultrasonic spray pyrolysis with hydrogen treatment, in this case, was used for the synthesis of titanium-based powders from titanium oxy-sulfate (TiOSO_4_ VENATOR Hombityl UN 3264, Duisburg, Germany). The innovation goal of this study is the testing of an industrial solution of TiOSO_4_ for the synthesis of nanosized TiO_2_, which is missing in the literature. For the experiments, industrial titanium oxy-sulfate was used at 1 M concentration, 99.9% purity, and 1361 g/cm^3^ density. An ultrasonic generator (PRIZNANO, Kragujevac, Serbia) was used for the production of aerosols. The process parameters which were observed are temperature, concentration of the precursor, and different hydrogen/argon ratios at a constant frequency of 1.7 MHz. Hydrogen and argon flow rates were controlled using a flowmeter and then mixed before entering the US generator. The process conditions for the experiments are outlined in Table 1.

All experiments were conducted at the IME, RWTH Aachen University. The methods used for sample characterization included the following:

X-ray diffraction (XRD): Powder XRD was performed with a Bruker D8 Advance diffractometer equipped with a LynxEye detector and a copper tube with a nickel filter (Cu Kα1.2 radiation, λ = 1.54187 Å). XRD patterns were collected using Bragg–Brentano geometry.

Scanning electron microscopy (SEM): SEM analysis was carried out using the JSM 7000F by JEOL (2006 model, JEOL Ltd., Tokyo, Japan), revealing an irregular structure of the precursor.

Energy-dispersive X-ray spectroscopy (EDX): EDX analysis was performed with an Octane Plus-A system by Ametek-EDAX (2015 model, AMETEK Inc., Berwyn, PA, USA) and analyzed using Genesis V 6.53 software by Ametek-EDAX.

For the analysis, HighScore software5.2 [31] with the COD database [32,33] was used.

In Figure 1, the equipment used in the experiments is shown.

### 2.1. Particle Size

The formation of particles will be initially defined by the diameter of an aerosol droplet (*D_ds_*), as indicated in Equation (1) [12,34]:(1)$$D_{ds}=0.34{\left(\frac{8\pi \cdot \sigma }{\rho \cdot f^{2}}\right)}^{\frac{1}{3}}$$
where *D_ds_* is the aerosol diameter (μm), *f* is the ultrasound frequency (MHz), *ρ* is the solution density (g/mL), and *σ* is the surface tension of the solution (N/m).

The particle size (*D_p_*) is influenced by both the droplet size and the concentration of the solution (*C_s_*). The relationship between the concentration and other precursor characteristics and the final particle size, assuming no precursor is lost in the process, can be described by Equation (2) [12,34]:(2)$$D_{p}=D_{ds}{\left(\frac{C_{s}\cdot M_{met}}{M_{p}\cdot \rho_{met}}\right)}^{\frac{1}{3}}$$
where *D_p_* is the particle diameter (nm), *D_ds_* is the aerosol droplet diameter (nm), *M_p_* is the molar mass of the precursor (g/mol), *ρ* is the density of particles (g/cm^3^), and *C_s_* is the concentration of the precursor solution (g/L).

Using Equations (1) and (2), droplet and particle size can be determined in theory, as shown in Figure 2.

Figure 2 illustrates that as the precursor concentration increases, the diameter of the aerosol droplet decreases. However, the particle size increases with higher precursor concentrations. For a concentration of 1 mol/L of TiOSO_4_ (160 g/L), the expected particle sizes for titanium and titanium oxide are approximately 590 nm and 720 nm, respectively. Additionally, an increase in the solution density leads to a reduction in droplet size, as indicated by the black line.

### 2.2. Theoretical Thermodynamic Analysis

The hydrogen reaction with titanium oxy-sulfate is shown by the reactions in Equations (4) and (5):(3)$$TiCl_{4}\left(g\right)+2H_{2}\left(g\right)=Ti+4HCl\left(g\right)$$
(4)$$TiOSO_{4}\left(ia\right)+H_{2}\left(g\right)=Ti+SO_{3}\left(g\right)+H_{2}O+0.5O_{2}\left(g\right)$$
(5)$$2TiOSO_{4}\left(ia\right)+2H_{2}\left(g\right)=TiO_{2}+Ti+2SO_{3}\left(g\right)+2H_{2}O$$
(6)$$TiO_{2}+2H_{2}\left(g\right)=Ti+2H_{2}O$$
(7)$$Ti+O_{2}\left(g\right)=TiO_{2}$$

There are different pathways for titanium oxy-sulfate decomposition. Below 650 °C, none of the reactions mentioned are likely to occur, as illustrated in Figure 3. Between 650 °C and 1160 °C, reaction (5) becomes thermodynamically feasible, allowing for the formation of both titanium dioxide and metallic titanium. At temperatures above 1150 °C, the Gibbs free energy of reaction (4) turns negative, making this reaction possible. However, reaction (3) demonstrates that titanium chloride cannot be directly reduced to metallic titanium under these conditions. Additionally, it is thermodynamically impossible to reduce titanium dioxide with hydrogen, as indicated by reaction (6). On the other hand, the oxidation of titanium to titanium oxide is the most favored reaction at room temperature following the negative values of Gibbs energy, as shown by Equation (7) and Figure 3.

## 3. Results and Discussion

### 3.1. Influence of Temperature

The effects of temperature on different parameters such as particle size, morphology and composition of the obtained titanium-based powders were investigated while other parameters were kept at constant values, as shown in Table 1.

EDS results for the samples at different temperatures are shown in Figure 4.

Judging by the EDS spectra, it is clearly visible that the products obtained under these conditions are mainly titanium dioxide by stoichiometry. The influence of temperature shows that sulfur content decreases as temperature increases, which can be explained by the incomplete decomposition of precursor titanium oxy-sulfate at lower temperatures. The peak at 1.5 keV is attributed to aluminum, which is the carrier material used for the sample.

SEM analysis was used along with EDS, as stated in Section 2, and the results are shown in Figure 5 for different temperatures.

As stated in the Introduction, using the USP method, spherical particles in the nanoscale range are obtained. The precursor solution is TiOSO_4_, which is decomposed at high temperature in a hydrogen atmosphere to titanium-based powders, mainly titanium dioxide.

### 3.2. Influence of Concentration

As described in Table 1, the concentration of the precursor solution was changed while other parameters remained constant (temperature 1000 °C, gas flow 1:1, and time 180 min). In Figure 6, the EDS spectra are shown for precursor concentration changes.

Analysis of the EDS spectra reveals that as the precursor solution is diluted, the titanium content decreases while the oxygen and sulfur contents increase. According to these data, the titanium-based powder is primarily titanium dioxide. The increase in sulfur and oxygen content with lower precursor concentrations can be attributed to the presence of incomplete transformation of the titanium oxy-sulfate in a hydrogen atmosphere. This occurrence may be due to the retention time in the reactor, which is influenced by the flow rate of the carrier gas. Similarly to the last EDS figure, the peak at 1.5 keV is attributed to the carrier material.

Particles obtained in these experiments were also characterized using SEM analysis (Figure 7).

Based on the results, the morphology of the particles obtained by varying the precursor concentration is clearly observable. Incomplete transformation of the titanium oxy-sulfate precursor is evident in the final case (Figure 7d), as previously discussed in relation to Figure 6, due to the presence of sulfur. Additionally, the sphericity of the resulting particles is distinctly visible, as confirmed and noted in the analysis.

### 3.3. Influence of Gas Flow

Gas flow influence was also examined while all other parameters remained constant. The first experiment in this part of the paper was performed without hydrogen in the reaction tube. Argon, as an inert gas, was used for the transportation of droplets (2 L/min) made in an ultrasonic spray generator. Two more experiments were carried out (Table 1) with a ratio of 2:1 (2 L/min of hydrogen and 1 L/min argon) and a 3:1 ratio (a ratio of 1:1 was already used in previous experiments). After the first experiment (experiment 9) (only argon), it can be concluded that the transformation of titanium oxy-sulfate is very slow or impossible without hydrogen. In the other two experiments (10 and 11), fine spherical nanoparticles were obtained.

The EDS spectra in Figure 8 reveal the elemental composition of oxygen (O), titanium (Ti), and sulfur (S) for samples with different gas flow ratios during synthesis. The results show that varying the gas flow ratio affects elemental distribution.

Sample (a), with a 3:1 ratio, exhibits the highest oxygen and lowest titanium content, indicating incomplete transformation due to the fastest gas flow and shortest residence time. Sulfur content is slightly higher than in (c).

Sample (b), with a 2:1 ratio, shows slightly higher oxygen and lower titanium compared to (c), reflecting reduced transformation efficiency with a shorter residence time. Sulfur content remains low.

Sample (c), with a 1:1 hydrogen-to-argon ratio, has higher titanium and lower oxygen content, indicating a more complete transformation of the titanium oxy-sulfate precursor due to longer residence time. Sulfur content is minimal, suggesting effective transformation and sulfur removal even at this lower flow rate.

In summary, lower gas flow ratios, providing longer residence times, yield higher titanium content, lower oxygen content, and minimal sulfur impurities, demonstrating their effectiveness in this synthesis method.

The SEM images (Figure 9) clearly confirm the formation of very fine titanium dioxide particles across all gas flow ratios. In image (a), corresponding to the highest gas flow ratio of 3:1, the particles are well formed, but with slight variation in morphology. As the gas flow ratio is reduced to 2:1, shown in image (b), the particles remain small and display greater uniformity. Finally, image (c), with the lowest gas flow ratio of 1:1, demonstrates the most consistent morphology, with particles appearing fine and well defined. These results confirm that regardless of the gas flow ratio, the synthesis process consistently produces very fine particles.

The results of the XRD analysis are shown in Figure 10.

The XRD results for particles obtained at 1000 °C, with a precursor concentration of 80 g/L and varying gas flow ratios, indicate that the dissolution of titanium oxy-sulfate is complete, resulting in the formation of titanium-based powders. Figure 10a,b shows similar results, forming both anatase and rutile. In Figure 10c, the titanium dioxide phase in the form of anatase is present, with clear interferences, suggesting that part of the product may be in an amorphous form and showing that 700 °C is insufficient for achieving a high-quality result.

### 3.4. Oxygen Concentration Analysis

In all experiments, the concentration of oxygen in the product was also measured using EDX quantitative analysis. The results of the experiments are shown in Figure 11.

The oxygen content in the samples obtained in this study aligns with that found in synthetic titanium dioxide, indicating that the primary product of the USP method is predominantly titanium dioxide. This finding is further supported by EDS spectrum analysis. The primary aim of this work was to assess the thermochemical prediction of forming titanium alongside titanium oxide (Equation (5)). Although Equation (5) suggests the formation of titanium, titanium’s strong affinity for oxygen leads to the formation of titanium oxide. Consequently, increasing the reaction temperature to 1450 °C is expected to favor titanium formation, which will be explored in future experiments. The presence of excess oxygen (higher than the stoichiometry value) may be due to surface oxidation of aluminum, which was used as a carrier for the sample.

## 4. Conclusions

In this work, the ultrasonic spray pyrolysis (USP) of titanium oxy-sulfate was successfully employed to synthesize titanium dioxide nanopowder. The key novelty lies in the use of titanium oxy-sulfate as a new precursor, showing its potential for producing high-purity titanium dioxide. Hydrogen flow during the process is essential, as the reaction is slow or incomplete without it. The study further revealed that low temperatures are insufficient to fully decompose the precursor, emphasizing the importance of adequate thermal conditions for successful synthesis.

The concentration of the precursor does not significantly impact the morphology of the final product, confirming that variations in concentration do not affect the formation of nanoscale titanium dioxide particles. Adjustments to the gas flow ratio were found to have an effect on particle morphology because it changes the residence time. These findings support the robustness of the USP method for producing consistent nanoscale titanium dioxide under varying conditions.

USP offers numerous advantages, including the ability to synthesize nanoparticles with high purity and homogeneity while minimizing contamination risks by avoiding mechanical processes like grinding. Additionally, the process is scalable and cost-effective, making it a viable method for large-scale industrial production of titanium dioxide nanopowder.

However, challenges such as energy consumption and handling the high surface area of the nanoparticles must be addressed. Future research could explore the possibility of producing metallic titanium under mild conditions, offering a more sustainable alternative to traditional smelting processes, as well as investigate the dependence of particle size on these parameters.

Titanium dioxide nanopowders produced via USP have a wide range of potential applications in photocatalysis, solar cells, sensors, biomedical devices, energy storage, catalysis, and coatings, owing to their high surface area and controlled crystal structures.

## Figures and Tables

**Figure 1 materials-17-04779-f001:**
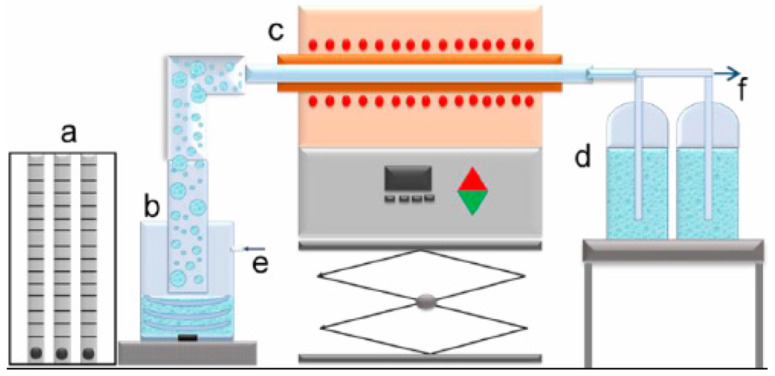
Scheme of USP equipment: (**a**) regulator for gas flow; (**b**) US generator; (**c**) furnace; (**d**) bottles for sample collection; (**e**) gas inlet, and (**f**) gas outlet.

**Figure 2 materials-17-04779-f002:**
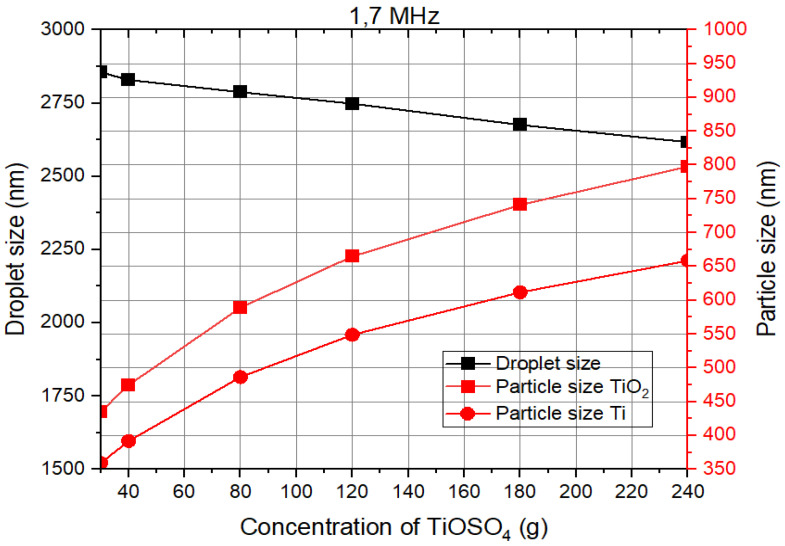
Droplet and particle size of the titanium powders in theory.

**Figure 3 materials-17-04779-f003:**
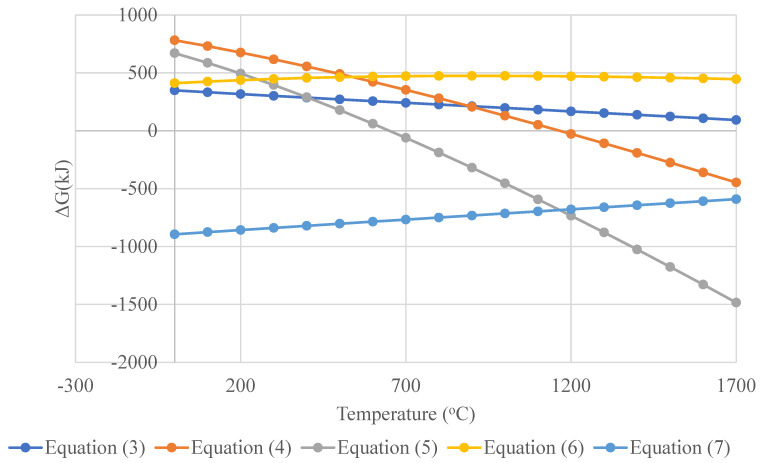
Gibbs free energy changes with temperature for the reactions of titanium chloride and titanium oxy-sulfate.

**Figure 4 materials-17-04779-f004:**
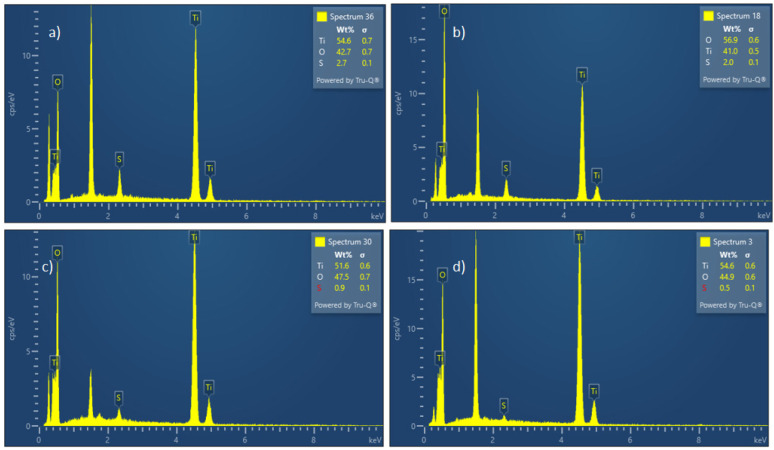
EDS diagrams for particles obtained at (**a**) 700 °C, (**b**) 800 °C, (**c**) 900 °C, and (**d**) 1000 °C.

**Figure 5 materials-17-04779-f005:**
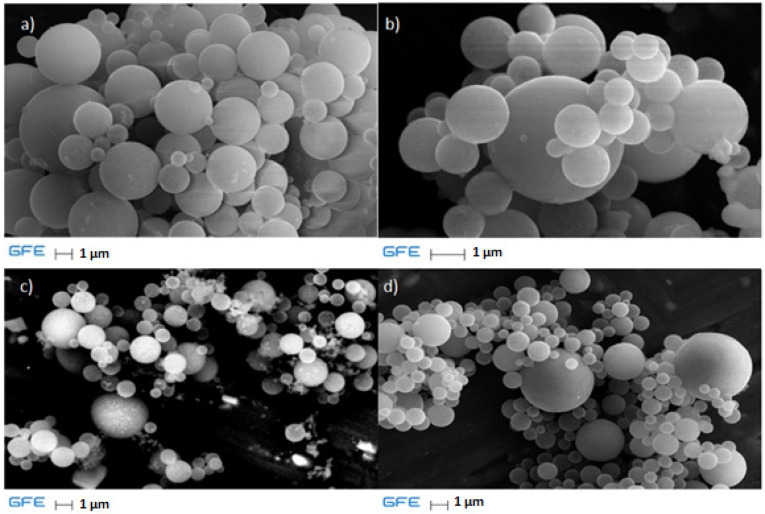
SEM analysis of samples for different temperatures: (**a**) 700 °C, (**b**) 800 °C, (**c**) 900 °C, and (**d**) 1000 °C.

**Figure 6 materials-17-04779-f006:**
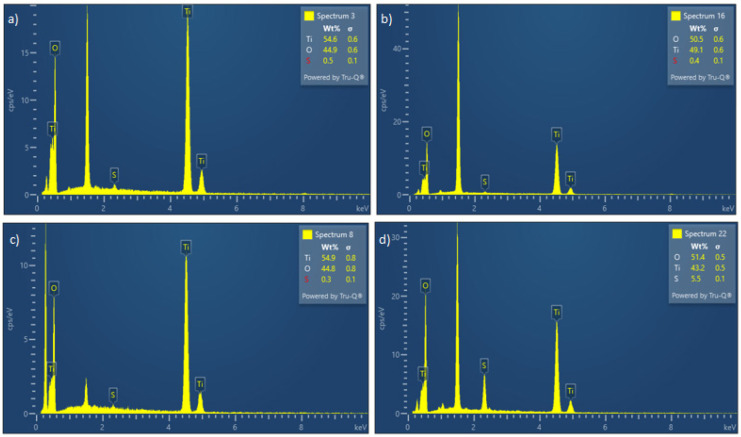
EDS spectra for particles obtained at precursor concentrations of (**a**) 80 g/L, (**b**) 60 g/L, (**c**) 48 g/L, and (**d**) 25 g/L.

**Figure 7 materials-17-04779-f007:**
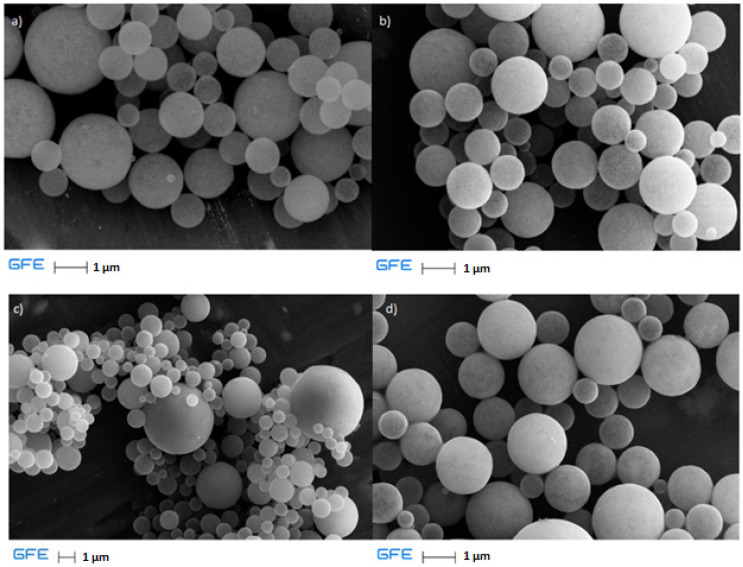
SEM analysis of samples for different concentrations: (**a**) 80 g/L, (**b**) 60 g/L, (**c**) 48 g/L, and (**d**) 25 g/L.

**Figure 8 materials-17-04779-f008:**
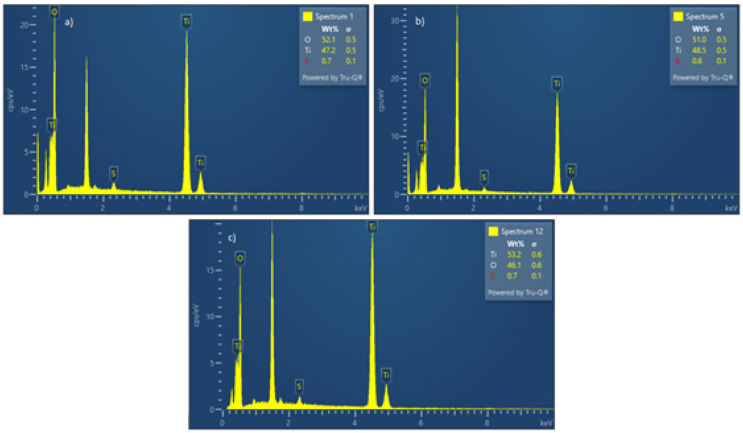
EDS spectra for particles obtained at gas flow ratios of (**a**) 3:1, (**b**) 2:1, and (**c**) 1:1.

**Figure 9 materials-17-04779-f009:**
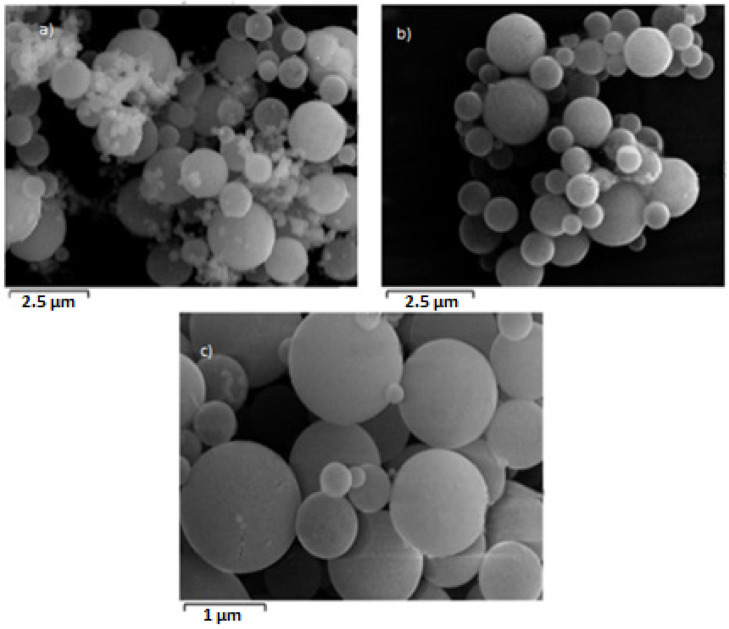
SEM analysis of samples for different gas flow ratios: (**a**) 3:1, (**b**) 2:1, and (**c**) 1:1.

**Figure 10 materials-17-04779-f010:**
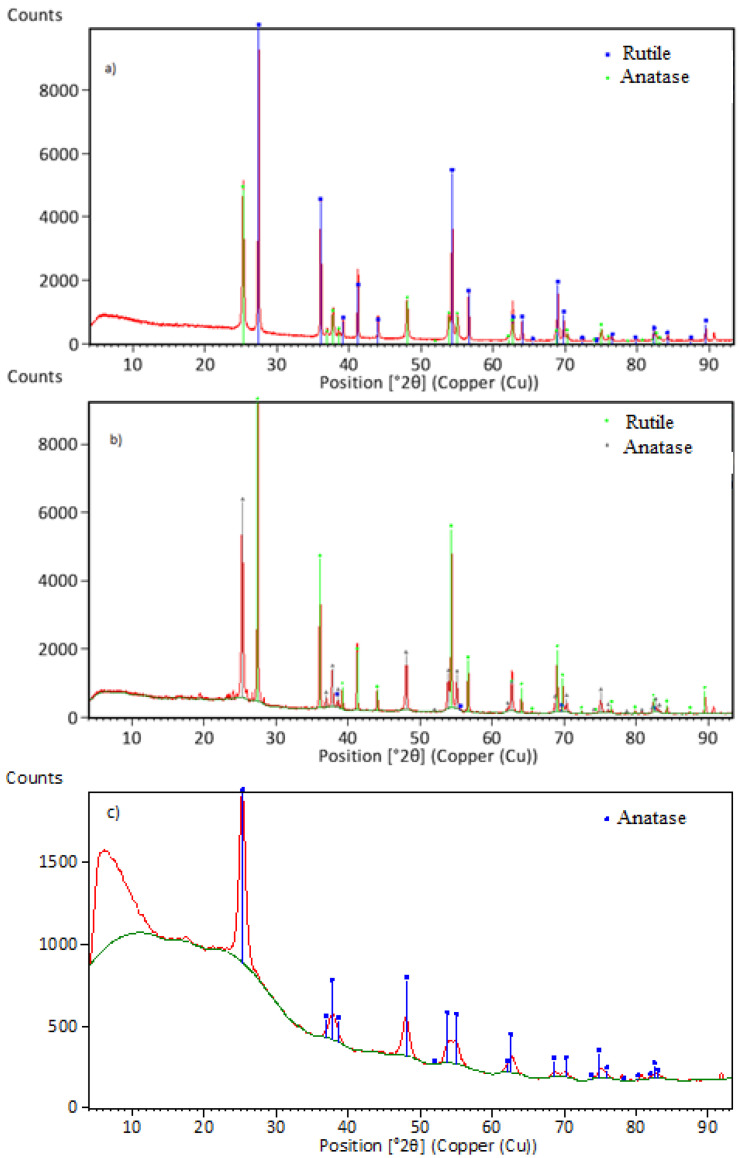
XRD results of particles at gas flow ratios of (**a**) 3:1 (1000 °C), (**b**) 1:1 (1000 °C), and (**c**) 1:1 (700 °C).

**Figure 11 materials-17-04779-f011:**
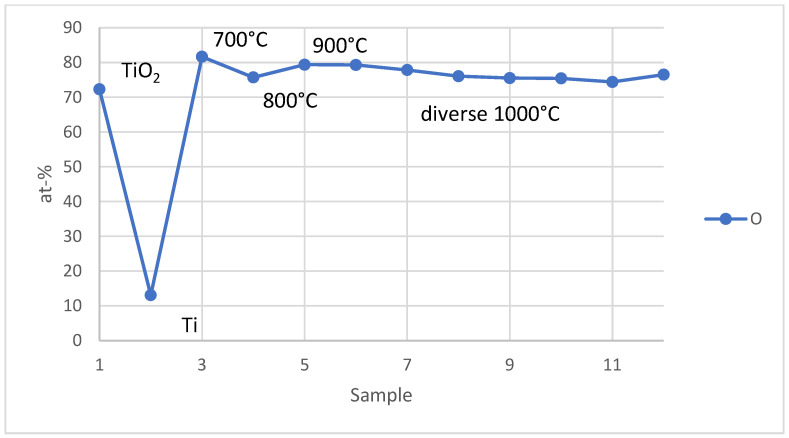
Oxygen concentration in synthetic titanium dioxide (sample 1), metal titanium (sample 2), and samples from experiments (samples 3–12).

**Table 1 materials-17-04779-t001:** Design of the experiments.

Exp. No.	Temperature, °C	Time, Min	Concentration, g/L	Solution Density, g/L	Flow Rate Ratio H2/Ar (L/min)
1	700	180	80	1.15	1:1
2	800	180	80	1.15	1:1
3	900	180	80	1.15	1:1
4	1000	180	240	1.39	1:1
5	1000	180	120	1.2	1:1
6	1000	180	80	1.15	1:1
7	1000	180	40	1.1	1:1
8	1000	180	80	1.15	Only argon
9	1000	180	80	1.15	2:1
10	1000	180	80	1.15	3:1

## Data Availability

The original contributions presented in the study are included in the article, further inquiries can be directed to the corresponding author.

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
