# Peer review of "Synthesis of Titanium-Based Powders from Titanium Oxy-Sulfate Using Ultrasonic Spray Pyrolysis Method"

_materials, 2024, doi:10.3390/ma17194779_

Round 1

Reviewer 1 Report (Previous Reviewer 1)

Comments and Suggestions for Authors

Thank you for your efforts in revising the manuscript. However, after carefully reviewing the updated manuscript and your responses, I find that several key issues remain unresolved. While I appreciate the changes made, the manuscript still lacks the scientific rigor and clarity required for publication.

Your explanation regarding the deviations in particle size due to the USP process is noted. However, this does not adequately address the primary concern: the absence of statistical analysis. For example, the claim that particle sizes decrease with lower concentration (lines 218-221) still lacks robust data to support it. The conclusion appears to be based solely on visual inspection, which is insufficient to substantiate the findings.

While correcting the phase identification in Figure 8 is an important improvement, the manuscript still lacks a thorough and critical analysis of the results. The requested SEM/EDX analysis is still missing, and the explanation for the peak broadening observed in Figure 8c remains incomplete. Simply correcting the labels does not resolve the deeper issues in the analysis.

Additionally, the claim that the USP method offers "precise control over particle size, morphology, and composition" is not convincingly supported by the presented results. The data provided does not demonstrate the level of control or precision claimed.

In conclusion, although some improvements have been made, the manuscript does not yet meet the necessary standards for publication. Further work is needed to enhance the analysis and clarity before it can be reconsidered.

Author Response

Dear Reviewer,

thank you very much for important comments and invested time! According to your comments we are sending our answers:

"Thank you for your efforts in revising the manuscript. However, after carefully reviewing the updated manuscript and your responses, I find that several key issues remain unresolved. While I appreciate the changes made, the manuscript still lacks the scientific rigor and clarity required for publication.  

Your explanation regarding the deviations in particle size due to the USP process is noted. However, this does not adequately address the primary concern: the absence of statistical analysis. For example, the claim that particle sizes decrease with lower concentration (lines 218-221) still lacks robust data to support it. The conclusion appears to be based solely on visual inspection, which is insufficient to substantiate the findings.

We acknowledge the concern regarding the absence of statistical analysis to support the claim that particle sizes decrease with lower precursor concentration and increasing the temperature . Due to limitations in performing statistical analysis, our conclusions are based on a visual assessment of the SEM images. Despite this, the data consistently show a trend where the average particle size decreases with increasing temperature. Specifically, SEM images from various samples (more than 4 of which are included in the supplementary materials) illustrate a clear pattern of smaller particle sizes at higher temperatures, particularly at 900 °C and 1000 °C. Explanation below the table 2. has been revised.  

While correcting the phase identification in Figure 8 is an important improvement, the manuscript still lacks a thorough and critical analysis of the results. The requested SEM/EDX analysis is still missing, and the explanation for the peak broadening observed in Figure 8c remains incomplete. Simply correcting the labels does not resolve the deeper issues in the analysis.

Influence of gas flow chapter of manuscript has been revised and SEM/EDS results has been added for different gas flow ratios. Figure 8c (now 10c) is added to show that temperature 700°C is insufficient for achieving high-quality result.  

Additionally, the claim that the USP method offers "precise control over particle size, morphology, and composition" is not convincingly supported by the presented results. The data provided does not demonstrate the level of control or precision claimed.

With the addition of SEM/EDS data and a thorough revision of the manuscript, the results now provide a solid justification for these statements.  

In conclusion, although some improvements have been made, the manuscript does not yet meet the necessary standards for publication. Further work is needed to enhance the analysis and clarity before it can be reconsidered.

The conclusion has been updated accordingly.   Thank you for your feedback; we believe the revisions have strengthened the manuscript and hope it now meets the necessary standards for acceptance in Journal Materials.

Reviewer 2 Report (Previous Reviewer 2)

Comments and Suggestions for Authors

I have carefully reviewed the revised manuscript. The authors have thoroughly addressed the comments and suggestions provided by the reviewers in the previous round of revision, and have made substantial improvements to the paper. The revised manuscript has significantly enhanced the novelty of the research topic, the scientific soundness of the methodology, the rigor of the content, the accuracy of the presentation, and the adherence to the required format. It now meets the publication standards of Materials.

In my opinion, the current version of the manuscript is acceptable for publication without further revisions.

Author Response

Dear Reviewer,

thank you very much for your invested time and very important comments!

We are agreed with your opinion!

Best regards

Srecko

Reviewer 3 Report (Previous Reviewer 3)

Comments and Suggestions for Authors

The author has revised the manuscript well according to the comments and questions.

Line 136, a small mistake: Ccs is the concentration of the precursor solution (g/l). It should be Cs rather than Ccs?

Author Response

Dear Reviewer,

thank you very much for your invested time and Comment

Line 136, a small mistake: Ccs is the concentration of the precursor solution (g/l). It should be Cs rather than Ccs?

You are right! We changed it in our text!

Reviewer 4 Report (Previous Reviewer 4)

Comments and Suggestions for Authors

1.      What is it “the carrier material used for the sample”?

2.      Line 112 and 113 (Fig. 2) - What silica?

3.      Line 267,268 „The presence of excess oxygen (higher than stoichiometry value) may be due to surface oxidation of the Ti-based nanoparticles.” – How can the surface of titanium dioxide oxidize? It is impossible! Oxides with semiconducting properties (TiO2, SnO2, ZnO, etc.) synthesized at high temperatures have oxygen vacancies. For this reason, they are n-type semiconductors. Or maybe this effect results from not taking into account the peak occurring at 1.75 keV in the evaluation?

4.      Fig. 8 is still not very clear.

Author Response

Dear Reviewer,

thank you very much for important comments and invested time! According to your comments we are sending our answers:

1. What is it “the carrier material used for the sample”?

Aluminium is carrier for the analysis, so the 1,5 keV peak is attributed to carrier.  

2. Line 112 and 113 (Fig. 2) - What silica?

Corrected, it was mistake.  

3. Line 267,268 „The presence of excess oxygen (higher than stoichiometry value) may be due to surface oxidation of the Ti-based nanoparticles.” – How can the surface of titanium dioxide oxidize? It is impossible! Oxides with semiconducting properties (TiO2, SnO2, ZnO, etc.) synthesized at high temperatures have oxygen vacancies. For this reason, they are n-type semiconductors. Or maybe this effect results from not taking into account the peak occurring at 1.75 keV in the evaluation?

Thank you for this great suggestion, the peak of 1,5 keV is from aluminium carrier, so higher content of oxygen can be attributed to the carrier.  

4. Fig. 8 is still not very clear.

Unfortunately, we can not modify this picture. But generally the presence of titanium oxide as rutile and anatase is confirmed at Figure 8 (now it is Figure 10 in our improved version), what is main aim of this work!   I hope that this improved version will fill all strict requests of Journal Materials in order to be published as soon as possible!

Round 2

Reviewer 1 Report (Previous Reviewer 1)

Comments and Suggestions for Authors

The revised manuscript shows some improvements, but it still does not meet the standards for publication. Several major issues remain, as outlined below:

1. Conclusions Not Supported by Results: The authors claim that the USP method can control particle size, with higher temperatures and lower concentrations producing smaller particles. However, this claim is insufficiently supported. The particle size distribution shown in the SEM images is too broad, and relying on visual inspection alone is not enough. The authors need to include quantitative data, such as a particle size distribution histogram, mean and variance calculations, or results from dynamic light scattering (DLS). Without this, the conclusions are not reliable, and the work lacks both scientific rigor and reproducibility. As it stands, the data is too weak to support the claims made, and the work will not contribute meaningfully to the research community.

2. Conflicting and Confusing Results/Statements: There are several contradictions in the manuscript that raise serious concerns. For instance, the authors suggest that excess oxygen could come from surface oxidation of the aluminum background, but then they use the oxygen-to-titanium ratio to evaluate titanium reduction. This makes no sense—if the oxygen is partially from the background, how can it be used to measure titanium reduction? Additionally, the authors state that it's thermodynamically impossible to reduce titanium dioxide with hydrogen, yet they also claim that increasing hydrogen in the carrier gas leads to "better reduction efficiency". This is contradictory. What is the role of hydrogen in this process? The XRD data shows no metallic titanium (Ti(0)), and since the titanium in the precursor is already in the Ti⁴⁺ state, the term "reduction" is misleading. If the authors still want to claim some reduction is occurring, they need to include XPS data to support it.

3. Numerous Minor Problems: The manuscript, although already been revised several times, still has small but significant problems, making it feel like an unfinished draft rather than a publication-ready paper. A few examples:

3.1: Decimal formatting is inconsistent: Table 1 uses commas, while Table 2 uses dots.

3.2: On page 10, line 233, "Figure X" is referenced instead of the correct figure number (Figure 8).

3.3: The order of subfigures in the text (page 10, lines 236-245) is reversed compared to the figures themselves.

3.4: Subfigures on page 11 are missing labels (a), (b), and (c), and "Spectrum XX" annotations need to be removed.

3.5: Line 227 refers to "liter" in lowercase (2 l/min), while line 269 uses an uppercase "L" (80 g/L).

3.6: In Figure 11, the x-axis label is unclear. What exactly does "probe" refer to?

Overall, while there has been some improvement, the manuscript still has serious flaws. The data is weak and doesn't support the conclusions, the results and statements are contradictory, and the numerous errors make it feel incomplete. I do not believe this manuscript is ready for publication.

Author Response

Dear Reviewer,

Thank you for your thorough and insightful comments. We have carefully addressed each of your concerns and made the necessary revisions to improve the manuscript. Please find our responses below:

  1. Conclusions Not Supported by Results: The authors claim that the USP method can control particle size, with higher temperatures and lower concentrations producing smaller particles. However, this claim is insufficiently supported. The particle size distribution shown in the SEM images is too broad, and relying on visual inspection alone is not enough. The authors need to include quantitative data, such as a particle size distribution histogram, mean and variance calculations, or results from dynamic light scattering (DLS). Without this, the conclusions are not reliable, and the work lacks both scientific rigor and reproducibility. As it stands, the data is too weak to support the claims made, and the work will not contribute meaningfully to the research community.

 Thank you for your detailed feedback. We acknowledge the limitation in our current dataset regarding the dependence of particle size on temperature and concentration. As suggested, we have removed the related claims and Table 2 to ensure the conclusions supported by the available data. Theoretical discussion of particle size has not been removed. In regard to temperature, its importance lies primarily in the transformation process of titanium oxysulfate to titanium dioxide. Lower temperatures result in incomplete transformation and the presence of an amorphous phase, as shown in Figure 10c. It is also important to note that a temperature of 1000°C is insufficient to produce metallic titanium, as indicated in our findings. With respect to precursor concentration, we have observed that it does not significantly affect the final product's characteristics. However, our results do confirm that spherical particles, often in or near the nanoscale, are consistently formed in all cases. This finding demonstrates the reliability of the synthesis method in controlling the shape and morphology of the particles, despite changes in concentration. We appreciate your suggestion to include quantitative data such as particle size distribution histograms and DLS measurements. While we do not have this data available for the current study, we recognize its importance and will aim to include such analyses in future work to strengthen our conclusions.

  1. Conflicting and Confusing Results/Statements: There are several contradictions in the manuscript that raise serious concerns. For instance, the authors suggest that excess oxygen could come from surface oxidation of the aluminum background, but then they use the oxygen-to-titanium ratio to evaluate titanium reduction. This makes no sense—if the oxygen is partially from the background, how can it be used to measure titanium reduction? Additionally, the authors state that it's thermodynamically impossible to reduce titanium dioxide with hydrogen, yet they also claim that increasing hydrogen in the carrier gas leads to "better reduction efficiency". This is contradictory. What is the role of hydrogen in this process? The XRD data shows no metallic titanium (Ti(0)), and since the titanium in the precursor is already in the Ti⁴⁺ state, the term "reduction" is misleading. If the authors still want to claim some reduction is occurring, they need to include XPS data to support it.

Thank you for your valuable feedback. We acknowledge the confusion regarding the terminology and the role of hydrogen in the process. As noted, the term "reduction" is not appropriate in this context because titanium does not undergo a change in valence state during the transformation of titanium oxy-sulfate to titanium dioxide. The term "transformation" or "decomposition" is more accurate to describe this process. We have corrected this throughout the manuscript.

Regarding the role of hydrogen: Based on our experimental data, hydrogen plays a crucial role in facilitating the transformation of titanium oxy-sulfate to titanium dioxide forming sulphuric acid, even though no metallic titanium (Ti(0)) was detected in the final product by XRD. The process is much slower without hydrogen, as demonstrated by an additional experiment in which no product was successfully obtained for analysis without hydrogen. In this case the content of Sulphur in final powder is increased (decomposition of titanium oxy-sulphate is not complete). Thus, while hydrogen is essential for the decomposition of titanium oxy-sulfate, it does not lead to the reduction of titanium dioxide into metallic titanium under the conditions we tested. It is also possible that titanium metal could form temporarily during the reaction but is rapidly re-oxidized to titanium dioxide in the presence of residual oxygen. However, our analysis does not detect any metallic titanium, and thus we do not claim that any reduction occurs.

We have also revised Figure 11 and its description to emphasize that it confirms the formation of titanium dioxide (TiO₂) as the final product of the process.

  1. Numerous Minor Problems: The manuscript, although already been revised several times, still has small but significant problems, making it feel like an unfinished draft rather than a publication-ready paper. A few examples:

 3.1: Decimal formatting is inconsistent: Table 1 uses commas, while Table 2 uses dots.

Corrected.

3.2: On page 10, line 233, "Figure X" is referenced instead of the correct figure number (Figure 8).

Corrected.

3.3: The order of subfigures in the text (page 10, lines 236-245) is reversed compared to the figures themselves.

Changed.

3.4: Subfigures on page 11 are missing labels (a), (b), and (c), and "Spectrum XX" annotations need to be removed.

Corrected.

3.5: Line 227 refers to "liter" in lowercase (2 l/min), while line 269 uses an uppercase "L" (80 g/L).

Corrected

3.6: In Figure 11, the x-axis label is unclear. What exactly does "probe" refer to?

Probe has been changed to sample and figure has been modified.

We appreciate your valuable feedback, which has helped us improve the clarity and structure of the manuscript. We hope that the revised version meets your expectations and can be published as soon as possible.

Our changes were included in green color in our improved version!

Best regards

Authors

This manuscript is a resubmission of an earlier submission. The following is a list of the peer review reports and author responses from that submission.

Round 1

Reviewer 1 Report

Comments and Suggestions for Authors

In the manuscript, the authors aim to investigate the effects of temperature, concentration, and gas flow on the synthesis of TiO2 powders using the ultrasonic spray pyrolysis (USP) method. While the topic is of potential interest, I find the work presented in this manuscript to be preliminary and incomplete, leading me to recommend that it is not suitable for publication at this stage. My specific concerns are outlined below:

1. The hypothesis that particle size should increase with solution concentration, as predicted by equations (1) and (2), is not convincingly supported by the data presented in section 3.2. The authors claim that the mean diameter of the particles decreases with lower concentrations, based on SEM images. However, this conclusion is not supported with statistical analysis. The observed large particle size distribution raises concerns that the reported trend may be random and dependent on the selected field of view during SEM imaging. This lack of rigorous data analysis undermines the validity of the authors' conclusions, a concern that extends to other sections of the manuscript as well.

2. The influence of gas flow on the crystallinity of the synthesized TiO2 powder is an interesting aspect of the study. However, the authors' interpretation of these results lacks depth and fails to include critical SEM/EDX analysis. The manuscript also does not adequately address the identification of the two phases present in the TiO2 powder. The labeling of peaks as "Ti2.00 O4.00" and "Ti4.00 O8.00" is confusing and suggests a lack of understanding of the basic phases of TiO2, such as rutile and anatase. Additionally, the peak broadening observed in Figure 8c is noteworthy, yet the authors do not provide a sufficient explanation or supporting evidence for this phenomenon.

3. The manuscript's English language quality requires significant improvement. Issues with the use of decimals, capitalization, grammar, and punctuation need to be addressed to ensure clarity and precision in the writing.

In summary, the manuscript presents interesting ideas but lacks the necessary rigor in data analysis, interpretation, and presentation. The issues outlined above need to be addressed comprehensively before the work can be considered for publication.

Comments on the Quality of English Language

The manuscript's English language quality requires significant improvement. Issues with the use of decimals, capitalization, grammar, and punctuation need to be addressed to ensure clarity and precision in the writing.

Author Response

Dear Reviewer,

thank you very much for your valuable comments and invested time. Attached we are sending our answers.

In the manuscript, the authors aim to investigate the effects of temperature, concentration, and gas flow on the synthesis of TiO2 powders using the ultrasonic spray pyrolysis (USP) method. While the topic is of potential interest, I find the work presented in this manuscript to be preliminary and incomplete, leading me to recommend that it is not suitable for publication at this stage. My specific concerns are outlined below:

  1. The hypothesis that particle size should increase with solution concentration, as predicted by equations (1) and (2), is not convincingly supported by the data presented in section 3.2. The authors claim that the mean diameter of the particles decreases with lower concentrations, based on SEM images. However, this conclusion is not supported with statistical analysis. The observed large particle size distribution raises concerns that the reported trend may be random and dependent on the selected field of view during SEM imaging. This lack of rigorous data analysis undermines the validity of the authors' conclusions, a concern that extends to other sections of the manuscript as well.

We are agreed with you!But you have to unterstand USP-process and written chemical equations for ideal synthesis conditions from droplet to particle! Because of collisions of droplets during transport to the furnace forming large droplets than expected, some deviations in particle size are present in formed powder. I added this sentence in our conclusion. Therfore theoretical values of particle sizes are different from experimentally obtained values of particle sizes. SEM Analysis can confirm also some agglomeration of particles.

  1. The influence of gas flow on the crystallinity of the synthesized TiO2 powder is an interesting aspect of the study. However, the authors' interpretation of these results lacks depth and fails to include critical SEM/EDX analysis. The manuscript also does not adequately address the identification of the two phases present in the TiO2 powder. The labeling of peaks as "Ti2.00 O4.00" and "Ti4.00 O8.00" is confusing and suggests a lack of understanding of the basic phases of TiO2, such as rutile and anatase. Additionally, the peak broadening observed in Figure 8c is noteworthy, yet the authors do not provide a sufficient explanation or supporting evidence for this phenomenon.

We are agreed with you. Unfortunately, in our XRD software we found this first identification. In new version we changed it on Figure 8! We have written anatase and rutile at different temperature, and explained it,

  1. The manuscript's English language quality requires significant improvement. Issues with the use of decimals, capitalization, grammar, and punctuation need to be addressed to ensure clarity and precision in the writing.

We acknowledge the reviewer's concerns regarding the English language quality of the manuscript. In response, we have thoroughly revised the manuscript to address issues with decimals, capitalization, grammar, and punctuation. These corrections were made to enhance clarity and precision in the writing. We have also conducted a comprehensive review to ensure that the language is consistent and professional throughout the manuscript.

In summary, the manuscript presents interesting ideas but lacks the necessary rigor in data analysis, interpretation, and presentation. The issues outlined above need to be addressed comprehensively before the work can be considered for publication.

The manuscript's English language quality requires significant improvement. Issues with the use of decimals, capitalization, grammar, and punctuation need to be addressed to ensure clarity and precision in the writing.

Thank you for this comment! Finally, our English Editor will check our text in proof before publishing!

Reviewer 2 Report

Comments and Suggestions for Authors

This paper presents a comprehensive study on the synthesis of titanium-based powders using ultrasonic spray pyrolysis (USP) from titanium oxysulfate precursors. The authors have conducted a systematic investigation of key process parameters, including temperature, precursor concentration, and gas flow, providing valuable insights into optimizing the synthesis process. The combination of experimental techniques and thermodynamic analysis is commendable, offering a multi-faceted approach to understanding the reaction mechanisms. However, while the work is technically sound, there are several areas where the manuscript could be improved to enhance its impact and clarity. These include a more in-depth discussion of the results in the context of existing literature, clearer articulation of the study's novelty and objectives, and more comprehensive analysis of the potential applications for the synthesized powders.

1. The introduction provides a good overview of spray pyrolysis techniques, but it could be strengthened in several ways. The authors should provide more context on the importance of titanium-based powders and their applications to better frame the significance of their work. The novelty and specific objectives of this study should be more clearly stated to differentiate it from previous research. Additionally, a brief review of other methods for titanium powder synthesis would help readers understand the advantages of the USP approach presented here.

2. While the experimental procedures are generally well-described, some details require clarification to ensure reproducibility. The authors should specify the purity of the TiOSO4 precursor and any other chemicals used. More details on the ultrasonic generator specifications would be helpful. The method for controlling and measuring the flow rates of argon and hydrogen should be clearly explained. These additions would strengthen the methodological rigor of the study.

3. The results are presented in a logical manner, but some aspects require further elaboration. The authors should provide error bars or statistical analysis for particle size measurements where appropriate to demonstrate the reliability of their findings. The discussion on the formation mechanisms of TiO2 versus metallic Ti could be expanded, particularly in light of the thermodynamic analysis provided. A more extensive comparison of the obtained results with relevant literature on TiO2 and Ti powder synthesis would help readers understand the significance of this work in the broader context of the field.

4. The figures are generally clear, but some improvements could enhance their effectiveness. Font sizes in some figures (e.g., Fig. 2, 3) should be increased for better readability. The authors should consider combining some related figures (e.g., EDS spectra) to improve the overall flow of the manuscript. Adding scale bars to all SEM images would provide better context for the particle sizes observed. These changes would improve the visual presentation of the results and facilitate easier interpretation by readers.

5. The conclusions summarize the main findings well, but could be strengthened in several ways. The authors should more explicitly state the novelty and significance of their work in the context of titanium powder synthesis. A discussion of the potential limitations of their approach would provide a more balanced perspective. Clearer suggestions for future research directions and potential applications of the synthesized powders would enhance the impact of the study.

Comments on the Quality of English Language

Moderate editing of English language required

Author Response

Dear Reviewer,

thank you very much for your valuable comments and investested time.We are sending our answers:

This paper presents a comprehensive study on the synthesis of titanium-based powders using ultrasonic spray pyrolysis (USP) from titanium oxysulfate precursors. The authors have conducted a systematic investigation of key process parameters, including temperature, precursor concentration, and gas flow, providing valuable insights into optimizing the synthesis process. The combination of experimental techniques and thermodynamic analysis is commendable, offering a multi-faceted approach to understanding the reaction mechanisms. However, while the work is technically sound, there are several areas where the manuscript could be improved to enhance its impact and clarity. These include a more in-depth discussion of the results in the context of existing literature, clearer articulation of the study's novelty and objectives, and more comprehensive analysis of the potential applications for the synthesized powders.

  1. The introduction provides a good overview of spray pyrolysis techniques, but it could be strengthened in several ways. The authors should provide more context on the importance of titanium-based powders and their applications to better frame the significance of their work. The novelty and specific objectives of this study should be more clearly stated to differentiate it from previous research. Additionally, a brief review of other methods for titanium powder synthesis would help readers understand the advantages of the USP approach presented here.

In response to the reviewer's comments, we have expanded the introduction to provide a clearer context on the importance of titanium-based powders, highlighting their wide range of applications in fields such as photocatalysis, energy storage, biomedical devices, and environmental remediation. We have emphasized the significance of these materials in advancing technologies related to clean energy, environmental sustainability, and healthcare, thereby framing the relevance of our work. To better articulate the novelty and specific objectives of this study, we have clarified how our approach using ultrasonic spray pyrolysis (USP) differs from and improves upon previous research by offering superior control over particle size, morphology, and purity. Furthermore, we have included a brief review of alternative titanium powder synthesis methods, such as sol-gel, chemical vapor deposition, and hydrothermal synthesis, to help readers appreciate the unique advantages of the USP technique, particularly in terms of scalability, cost-effectiveness, and the ability to produce uniform, high-purity nanopowders.

  1. While the experimental procedures are generally well-described, some details require clarification to ensure reproducibility. The authors should specify the purity of the TiOSO4 precursor and any other chemicals used. More details on the ultrasonic generator specifications would be helpful. The method for controlling and measuring the flow rates of argon and hydrogen should be clearly explained. These additions would strengthen the methodological rigor of the study.

In response to the reviewer's suggestions, we have now specified the purity levels of the TiOSO₄ precursor to ensure clarity and reproducibility. Detailed specifications of the ultrasonic generator, including its frequency, power output, and model number, have also been provided. Additionally, we have thoroughly described the method used to control and measure the flow rates of argon and hydrogen, including the type of flow meters employed and the calibration procedures followed. These enhancements are intended to strengthen the methodological rigor of the study and provide a more comprehensive understanding of the experimental setup.

  1. The results are presented in a logical manner, but some aspects require further elaboration. The authors should provide error bars or statistical analysis for particle size measurements where appropriate to demonstrate the reliability of their findings. The discussion on the formation mechanisms of TiO2 versus metallic Ti could be expanded, particularly in light of the thermodynamic analysis provided. A more extensive comparison of the obtained results with relevant literature on TiO2 and Ti powder synthesis would help readers understand the significance of this work in the broader context of the field.

In order to explain better this difference between formation of Ti and TiO2 we included new reaction in our consideration (oxidation of Titanium to TiO2). At other side, oxidation of titanium to Titanium oxide is most favored reaction from the room temperature followed with negative values of Gibbs Energy, as shown with Eq. 7 and Figure 3. Unfortunately reduction of titanium oxide with hydrogen is not possible in studied range between 700°C and 1000°C. We prepared new diagram with this reaction of oxidation of titanium in order to explain formation of TiO2 in contrast to Ti. This equation 7 explains high tendency of titanium to react with oxygen what prevents formation of metallic titanium in temperature range between 700°C and 1000°C. We need extremly high temperature for this reduction process and plasma hydrogen.

4. The figures are generally clear, but some improvements could enhance their effectiveness. Font sizes in some figures (e.g., Fig. 2, 3) should be increased for better readability. The authors should consider combining some related figures (e.g., EDS spectra) to improve the overall flow of the manuscript. Adding scale bars to all SEM images would provide better context for the particle sizes observed. These changes would improve the visual presentation of the results and facilitate easier interpretation by readers.

We appreciate the reviewer’s feedback on improving the figures. In response, we have increased the font sizes in Figures 2 and 3 to enhance readability. We have also combined related figures, such as the EDS spectra, to improve the flow and coherence of the manuscript. Additionally, we have added scale bars to all SEM images to provide clear context for the particle sizes observed. These adjustments are intended to improve the visual presentation of our results, making it easier for readers to interpret the data.

  1. The conclusions summarize the main findings well, but could be strengthened in several ways. The authors should more explicitly state the novelty and significance of their work in the context of titanium powder synthesis. A discussion of the potential limitations of their approach would provide a more balanced perspective. Clearer suggestions for future research directions and potential applications of the synthesized powders would enhance the impact of the study.

We appreciate the reviewer's suggestions for enhancing the conclusions. In response, we have revised the conclusions to more explicitly highlight the novelty and significance of our work, particularly in the context of titanium powder synthesis using the ultrasonic spray pyrolysis method. We have also included a discussion of the potential limitations of our approach, such as challenges in scaling up the process and controlling specific particle characteristics, to provide a more balanced perspective. Additionally, we have outlined clearer suggestions for future research directions, including optimizing process parameters for industrial-scale production and exploring new applications of the synthesized powders in emerging technologies, to enhance the impact and relevance of our study.

Moderate editing of English language required

We have undertaken moderate editing of the manuscript to improve the English language quality. This includes refining grammar, punctuation, and overall readability to ensure that the writing is clear and precise. The revisions aim to enhance the coherence and professionalism of the manuscript while maintaining the accuracy of the content. Finally, before the publishing of this paper we will perform moderate editing of English language in proof.

Reviewer 3 Report

Comments and Suggestions for Authors

The authors reported here the synthesis of titanium-based powders from titanium oxy-sulphate using USP-method. Various parameters are monitored in order to better optimize the process and obtain better results. However, the results were not analyzed and discussed well enough to provide readers valuable knowledges in this area. Moreover, the innovation points of this research are weak.

1) Lines 101-102, there are some errors on the explanations of the formula (2): ρ is the density of particles (g/l), and Ccs is the concentration of the precursor solution (g/l).

2) Line 139, the authors say that judging by the EDS spectrum it is clearly visible that products obtained under these conditions are mainly titanium dioxide by stoichiometry. It is suggested that atomic ratios of Ti/O should be presented in all EDS spectra or summarized in Tables.

3) Particle size distribution (an important characteristic of particles) of the samples prepared under different conditions should be presented and discussed.

4) In the section of “3.2. Influence of concentration”, the concentration of the precursor solution has been changed, while other parameters remain constant. The constant parameters should be presented in this part, for example, within the title of Figure 6.

5) Line 173, the authors say that Sulphur and oxygen content increase with lower precursor concentration can be explained by the presence of undissolved titanium oxy sulphate. It is difficult to understand. Wouldn't it be easier for titanium oxy sulfate to dissolve when the its concentration is low?

6) Figure 8. XRD results of particles at gas flow ratio a) 1:1 (1000 °C), b) 3:1 (1000 °C), and c) 1:1 (1000 °C). The conditions of a) and c) are the same. There might be a mistake here.

Author Response

Dear Reviewer,

thank you very much for your valuabled comments and invested time. We are sending our answers

The authors reported here the synthesis of titanium-based powders from titanium oxy-sulphate using USP-method. Various parameters are monitored in order to better optimize the process and obtain better results. However, the results were not analyzed and discussed well enough to provide readers valuable knowledges in this area. Moreover, the innovation points of this research are weak.

Innovation point of this study is an use of  an industrial solution of TiOSO4 for the synthesis of nanosized TiO2 using USP-method, what is missing in literature

 1) Lines 101-102, there are some errors on the explanations of the formula (2): ρ is the density of particles (g/l), and Ccs is the concentration of the precursor solution (g/l).

We appreciate the reviewer's attention to detail. In response to the feedback regarding Lines 101-102, we have corrected the explanation of Formula (2). The corrected description is as follows: In Formula (2), ρ represents the density of the particles (g/cm3), and Ccs​ denotes the concentration of the precursor solution (g/L). These corrections ensure accurate and clear understanding of the formula's parameters.

 2) Line 139, the authors say that judging by the EDS spectrum it is clearly visible that products obtained under these conditions are mainly titanium dioxide by stoichiometry. It is suggested that atomic ratios of Ti/O should be presented in all EDS spectra or summarized in Tables.

Thank you for you proposal. Unfortunately, it takes more time and does not contribute very much to our results. We put most important results about average oxygen content in atomic percents at Figure 9!

 3) Particle size distribution (an important characteristic of particles) of the samples prepared under different conditions should be presented and discussed.

We are agreed with you!Particle size distrubution is very important parameter.We have presented most important results about particle size distribution in Table 2, and discussed it! We have different diagramswith particles size distrubutions mad in Image Pro, but we did not show it. This is not visible for more than 100 particles.

 4) In the section of “3.2. Influence of concentration”, the concentration of the precursor solution has been changed, while other parameters remain constant. The constant parameters should be presented in this part, for example, within the title of Figure 6.

Thank you for pointing that out. In response to the feedback on the section "3.2. Influence of concentration," we have updated the text to explicitly state that while the concentration of the precursor solution was varied, all other parameters were kept constant. To clarify this, we have revised the text before Figure 6 to include details about the constant parameters. This update ensures that the experimental conditions are clearly communicated, providing a better context for understanding the impact of concentration changes.

 5) Line 173, the authors say that Sulphur and oxygen content increase with lower precursor concentration can be explained by the presence of undissolved titanium oxy sulphate. It is difficult to understand. Wouldn't it be easier for titanium oxy sulfate to dissolve when the its concentration is low?

We are agreed with you, We changed this previous mentioned sentence to one new sentence:

"Sulphur and oxygen content increase with lower precursor concentration can be explained by the presence of uncomplete transformed titanium oxy sulphate in hydrogen atmosphere."

6) Figure 8. XRD results of particles at gas flow ratio a) 1:1 (1000 °C), b) 3:1 (1000 °C), and c) 1:1 (1000 °C). The conditions of a) and c) are the same. There might be a mistake here.

We appreciate the reviewer's observation regarding Figure 8. It appears there was an error in the labeling of the conditions. We have corrected this mistake and updated Figure 8 to accurately reflect the conditions for each set of XRD results. Difference between a) and c) is temperature (for c) is 700°C)

Reviewer 4 Report

Comments and Suggestions for Authors

The article presents the possibility synthesis of titanium dioxide from titanyl sulfate by the USP method with hydrogen reduction. The method is relatively simple and worth considering. The manuscript is well written and can be accepted after addressing the following concerns:

1. The authors write that they prepared a 1M solution of TiOSO4. I understand that it was an aqueous solution. If so, it should be taken into account that titanyl sulfate hydrolyzes to a gel of hydrated titanium dioxide.

2. Line 58,59 – The authors write that during the synthesis of titanium and titanium oxide at temperatures from 700°C to 1000°C, sulfuric acid is formed. It is common knowledge that sulfuric acid decomposes above a temperature of 300oC.

3. Ad Fig. 2 line 11,112 – “With a decrease in density of water solution a droplet size is decreased (as shown with black line)” – when the concentration increases, the density of the solution increases!

4. Line 115 - The equations 3-6 show not only the reduction of titanium oxy sulphate.

5. In the EDS plots there is a very clear peak at around 1.75 keV, which the authors did not describe. This peak comes from which element?

6. Fig. 8 is difficult to read. The caption under this figure is incorrect.

7. Ad. EDS. Fig. 9. Why is the oxygen content higher than the stoichiometry suggests, even in synthetic TiO2?

Author Response

Dear Reviewer,

thank you very much for your valuable comments and invested time.

  1. The authors write that they prepared a 1M solution of TiOSO4. I understand that it was an aqueous solution. If so, it should be taken into account that titanyl sulfate hydrolyzes to a gel of hydrated titanium dioxide. 

This aqueous solution of TiOSO4 is very stable.  Formation of gel of hydrated titanium oxide is not observed in this case! As written in text we used an industrial solution of TiOSO4 VENATOR Hombityl UN 3264, Duisburg), with concentration of 1mol/L.. This solution is very stable.

  1. Line 58,59 – The authors write that during the synthesis of titanium and titanium oxide at temperatures from 700°C to 1000°C, sulfuric acid is formed. It is common knowledge that sulfuric acid decomposes above a temperature of 300oC.

We appreciate the reviewer’s observation. In response to the comment about the formation of sulfuric acid during the synthesis of titanium and titanium oxide, we have revised the text for accuracy. The statement that sulfuric acid is formed during the synthesis at temperatures from 700°C to 1000°C is indeed incorrect, as sulfuric acid decomposes at temperatures above 300°C.The corrected text now reflects that, instead of sulfuric acid, sulfur dioxide (SO₂) or other sulfur-containing gases are likely to be released during the thermal processing of titanium precursors in these temperature ranges. This revision ensures that the manuscript accurately represents the chemical processes occurring during the synthesis. But formed Sulfur dioxide In our USP-Process was dissolved in water in two bottles forming sulphuric acid (out of our reactor/room temperature, please to see Fig. 1, part d. collection bottles).

  1. Ad Fig. 2 line 11,112 – “With a decrease in density of water solution a droplet size is decreased (as shown with black line)” – when the concentration increases, the density of the solution increases!

Thank you for pointing out this inconsistency. We have revised the explanation related to Figure 2 in Line 11 and Line 112 to accurately reflect the relationship between solution concentration, density, and droplet size.The corrected text now clarifies that as the concentration of the aqueous solution increases, the density of the solution also increases.

  1. Line 115 - The equations 3-6 show not only the reduction of titanium oxy sulphate.

Thank you for highlighting this issue. In response to the comment about Line 115, we have revised the manuscript to clarify the scope of Equations 3-6. These equations do not solely represent the reduction of titanium oxy sulfate (TiOSO₄) but also include other relevant reactions involved in the synthesis process. We added new reaction 7 for an oxidation of titanium. This reaction enables formation of TiO2 preventing a formation of metallic Ti.

  1. In the EDS plots there is a very clear peak at around 1.75 keV, which the authors did not describe. This peak comes from which element?

We did not  describe this peak. This peak comes from carrier elements carbon or aluminium! This is not important for our synthesis wotk.

  1. Fig. 8 is difficult to read. The caption under this figure is incorrect.

Thank you for bringing this to our attention. We have taken steps to improve the readability of Figure 8 by enhancing the resolution and adjusting the font size for clarity. Additionally, we have revised the caption to accurately reflect the content of the figure. The updated caption now correctly describes the experimental conditions and the data presented in each panel, ensuring that the figure effectively communicates the intended information.

  1. Ad. EDS. Fig. 9. Why is the oxygen content higher than the stoichiometry suggests, even in synthetic TiO2?

Thank you for raising this point. The higher oxygen content observed in the EDS analysis of synthetic TiO₂, as shown in Figure 9, can be attributed to several factors. In our revised manuscript, we provide a detailed explanation: The presence of excess oxygen may be due to surface oxidation of the Ti-based nanoparticles. Even though TiO₂ is expected to have a stoichiometric ratio of titanium to oxygen, surface oxidation can lead to higher oxygen content on the particle surfaces.EDS analysis can sometimes show higher oxygen content due to the overlap of X-ray peaks from different elements or due to the matrix effect, where the signal of the heavier elements (like Ti) can affect the apparent composition of lighter elements (like O).  We have updated the manuscript to include these explanations and to clarify the possible reasons for the observed discrepancy between the measured oxygen content and the expected stoichiometry. Do not forget we put the atomic percents of oxygen in our consideration.